# LOG-NORMAL STATE-SPACE MODEL

## ABSTRACT

State space model (SSM) has emerged as a strong alternative to transformer owing to its linear-time complexity and state retention mechanism where the computation efficiency and memory capability are enhanced especially in long-sequence tasks. However, the features derived from state updates in SSM still exhibit weaker representation than those generated by self-attention in transformer. This paper presents a new architecture that preserves the linear-time efficiency of SSMs while enabling state-update features to attain the expressiveness of self-attention, thereby achieving both computation efficiency and memory enhancement. In the experiments, the state-update mechanism of our SSM demonstrates superior performance compared to the other methods. On long-sequence tasks, our approach not only exhibits stronger long-range dependency modeling but also requires fewer computational resources than self-attention in transformers . Our code is available at https://anonymous.4open.science/r/Log-Normal-State-Space-Model-8301/.gitignore

## 1 INTRODUCTION

Modern large language models, such as GPT (Radford et al., 2019) and LLaMA (Touvron et al., 2023), are primarily built upon the transformer (Vaswani et al., 2017) architecture. A standard transformer block consists of two components including the self-attention (Vaswani et al., 2017), which captures the correlation among individual tokens within a sequence, and the multilayer perceptron (MLP), which aggregates the information across feature dimensions. Self-attention plays an important role behind the strong contextual modeling in transformer. However, transformer suffers from scaling problem where a quadratic time complexity is required in self-attention as the correlations between individual pairs of tokens in a sequence must be computed. Furthermore, due to the lack of an inherent memory mechanism, transformer requires a full temporal context to be reprocessed at every time step during generation, and results in increasing computational cost as the sequence length scales up. To relax this limitation, linear transformer (Katharopoulos et al.,

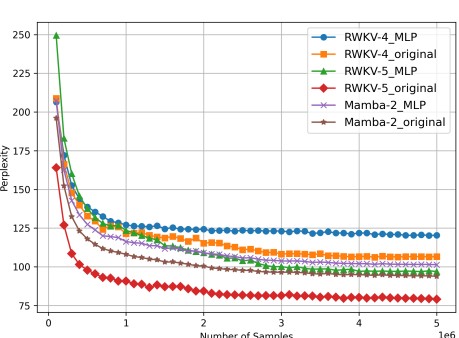

Figure 1: Training curves of the perplexity of validation data by using original SSM and SSM with channel-mixing layer replaced by a simple MLP (Radford et al., 2019). Models are trained on SlimPajama-1B Shen et al. (2023) under identical settings. These curves compare the learning process by SSMs based on RWKV and Mamba.

2020; Peng et al., 2021; Choromanski et al., 2021; Wang et al., 2020) was proposed to implement a transformer with linear time complexity. Another line of researches explore the state-space model (SSM) (Gu et al., 2022), which resembles recurrent neural network (RNN) (Schmidt, 2019) as a state retention machine, carrying past information forward through a hidden state. In addition, SSM is feasible to resemble the calculation of self-attention in transformer, and results in the emerging machines such as Mamba (Gu & Dao, 2023; Dao & Gu, 2024) and the receptance-weighted key value (RWKV) (Peng et al., 2023; 2024). SSM has demonstrated superior performance compared to transformer in long-context modeling. However, the model effectiveness relies not only on the core time-mixing layers but also on the specialized channel-mixing layers.

As shown in Figure 1, when the time-mixing layer of an SSM (either RWKV or Mamba) is retained but its original channel-mixing layer is replaced by a GPT-2's MLP (Radford et al., 2019), the result-

ing performance degrades significantly. This observation indicates that the desirable performance of SSM originates not solely from time-mixing mechanism. Channel-mixing layers also play a critical role. Such a reliance on the intricate design of the channel-mixing reduces the interpretability of a model while the added complexity increases both parameter size and computational cost. This study presents the log-normal state-space model (LNSSM) aiming at a new model that achieves the performance of time-mixing comparable to that of self-attention without relying on the specially-designed channel-mixing scheme while at the same time retaining the computation efficiency as a benefit from the variant of state-space model.

## 2 STATE-SPACE MODEL AND LINEAR TRANSFORMER

The recent emerging state-space model has increasingly empha-sized its structural similarity or equiva-lence to transformer, often showcasing even stronger ex-pressive capability. For example, Mamba Gu & Dao (2023) introduces a selective mechanism to classi-cal SSM, making the transition dynamics of hidden state $\boldsymbol{h}$ to be input dependent, as given in

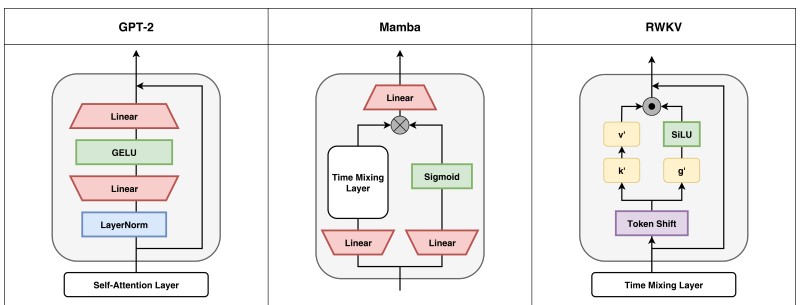

Figure 2: Channel-mixings (with colors) in GPT-2 (via MLP), Mamba and RWKV.

$$\boldsymbol{h}_t = A_t \boldsymbol{h}_{t-1} + B_t \boldsymbol{x}_t, \quad \boldsymbol{y}_t = C_t \boldsymbol{h}_t \tag{1}$$

where $A_t$, $B_t$ and $C_t$ denote the state, input and output matrices, respectively. In the empirical evaluations, Mamba (Gu & Dao, 2023) has demonstrated substantial potential to replace transformer in various tasks. Its enhanced variant, Mamba-2 (Dao & Gu, 2024), moves a step further by revealing a duality between SSM and attention mechanism, suggesting that two calculations can be considered functionally equivalent under certain conditions.

RWKV-4 (Peng et al., 2023) works as a type of SSM, but closely resembles its computation pattern as that of transformer where the output $\boldsymbol{y}_t$ at time step $t$ in a sequence length $N$ is expressed by

$$\boldsymbol{y}_t = \sigma(\boldsymbol{r}_t) \odot \left( \frac{\sum_{j=1}^{t-1} e^{-(t-1-j)\boldsymbol{w}+\boldsymbol{k}_j} \odot \boldsymbol{v}_j + e^{\boldsymbol{u}+\boldsymbol{k}_t} \odot \boldsymbol{v}_t}{\sum_{l=1}^{t-1} e^{-(t-1-l)\boldsymbol{w}+\boldsymbol{k}_l} + e^{\boldsymbol{u}+\boldsymbol{k}_t}} \right) \tag{2}$$

where $\{\boldsymbol{r}_t, \{\boldsymbol{w}, \boldsymbol{u}\}, \boldsymbol{k}_t, \boldsymbol{v}_t\}$ is the receptance, weight decay, key and value (RWKV), $\sigma(\cdot)$ is the sigmoid and $\odot$ is the element-wise product. Its subsequent versions, RWKV-5 and 6 (Peng et al., 2024), continue to adopt key-value formulation that acts as a style of self-attention architecture via

$$\boldsymbol{y}_t = \boldsymbol{r}_t \odot \left( \sum_{l=1}^{t-1} (e^{\boldsymbol{w}})^{t-1-l} \odot \boldsymbol{k}_l^\top \boldsymbol{v}_l + \boldsymbol{u} \odot \boldsymbol{k}_t^\top \boldsymbol{v}_t \right). \tag{3}$$

Despite the success of using SSMs, the performance of these methods heavily rely on the crafted network architecture. As shown in Figure 2, Mamba incorporates the customized skip connections with linear projection. RWKV treats the channel-mixing layer as an attention-like mechanism with token-shift operations. Accordingly, the merit of using Mamba and RWKV is not solely caused by the time-mixing layer, which handles the temporal dependencies, but also by the specially designed channel-mixing (or MLP) components that significantly improve the overall performance.

Although SSM and self-attention methods are theoretically related in the sense that both methods characterize the token dependencies in a text string, their computational behaviors and distributional characteristics differ substantially. In particular, the additive accumulation process in SSM does not exhibit the benefits of dynamic weighting behavior or output distribution similar to what self-attention does. As a result, despite the theoretical connection between SSM and self-attention, SSM cannot be directly compared with self-attention in terms of practical performance.

On the other hand, linear transformer aims to reduce the quadratic time complexity of self-attention mechanism by reformulating the attention as a kernel-based operation. The core idea is to approximate the softmax attention function using kernel methods, where the attention kernel $\kappa(\boldsymbol{q}, \boldsymbol{k})$ is defined as the inner product of transformed query and key vectors through a feature map $\Phi(\cdot)$. Therefore, the attended feature embedding using linear transformer is calculated by

$$\boldsymbol{y}_t = \frac{\sum_{j=1}^{N} \Phi(\boldsymbol{q}_t)^\top \Phi(\boldsymbol{k}_j) \boldsymbol{v}_j^\top}{\sum_{l=1}^{N} \Phi(\boldsymbol{q}_t)^\top \Phi(\boldsymbol{k}_l)}. \tag{4}$$

Here, the inner product $\langle \Phi(\boldsymbol{q}_t), \Phi(\boldsymbol{k}_j) \rangle$ is used to approximate the softmax kernel $e^{\boldsymbol{q}_t^\top \boldsymbol{k}_j}$. Equation (4) highlights an important computation property, namely, for a fixed query token $\boldsymbol{q}_t$, the summation over keys and values is independent of $t$ and can thus be pre-aggregated. This leads to the calculation of the reformulated linear-time feature embedding as

$$\boldsymbol{y}_t = \frac{\Phi(\boldsymbol{q}_t)^\top \sum_{j=1}^{N} \Phi(\boldsymbol{k}_j) \boldsymbol{v}_j^\top}{\Phi(\boldsymbol{q}_t)^\top \sum_{l=1}^{N} \Phi(\boldsymbol{k}_l)}. \tag{5}$$

This formulation enables changing the order of matrix multiplications among query, key, and value components, allowing for linear time complexity in sequence length. This is achieved without additional memory cost, unlike the cache-based method such as KV cache (Pope et al., 2023).

However, this efficiency comes at the expense of decreasing the expressiveness. Since the attention is no longer computed through direct pairwise interactions between individual tokens, as provided in the softmax attention, this model may suffer from the degraded performance in practice. The change of attention computation may alter the core dynamics of query-key-value (QKV) interactions, which will lead to a loss in the model's ability to capture the fine-grained token relationships.

# 3 IMPROVING THE PERFORMANCE OF LINEAR ATTENTION

## 3.1 LINEAR LOG-NORMAL ATTENTION

A recent study introduced a variant of linear attention called linear log-normal attention (LLN) (Nahshan et al., 2023), aiming to bridge the performance gap between linear attention and softmax attention. LLN was begun by analyzing the statistical property of attention weights in traditional transformer and theoretically demonstrated that self-attention exhibited a log-normal distribution under certain conditions. The output of the softmax attention could be approximated by a log-normal distribution under mild assumptions. In particular, if the $d$-dimensional queries $\boldsymbol{q}_i$ and keys $\boldsymbol{k}_j$ are assumed to be zero-mean isotropic-variance Gaussian, i.e. $\boldsymbol{q}_i \sim \mathcal{N}(0, \sigma_q^2 I)$ and $\boldsymbol{k}_j \sim \mathcal{N}(0, \sigma_k^2 I)$, then the dot-product attention score $a_{ij} = (\boldsymbol{q}_i^\top \boldsymbol{k}_j)/\sqrt{d}$ becomes approximately Gaussian due to the central limit theorem (Lee et al., 2018). The softmax attention weights are defined as

$$w_{ij} = \frac{\exp(a_{ij}/\tau)}{\sum_{l=1}^{N} \exp(a_{il}/\tau)} \tag{6}$$

where $\tau$ is a temperature controlled by the variance of $a_{ij}$. Since both the numerator and denominator in the softmax function involve exponential of a Gaussian, the softmax attention $w_{ij}$ can be approximated as a log-normal random variable. By applying the theorem in (Fenton, 1960), which states that the sum of log-normal variables can itself be approximated as a log-normal, the resulting ratio in $w_{ij}$ also approximately follows a log-normal distribution with parameters $\mu_m$ and $\sigma_m^2$ as

$$w_{ij} \sim \mathrm{LogNormal}(\mu_w, \sigma_w^2), \quad \mu_w = -\ln N - \frac{1}{2}\sigma_w^2, \quad \sigma_w^2 = \sigma_q^2 \sigma_k^2 + \sigma_{qk}^2 \tag{7}$$

where $\sigma_{qk}^2$ captures the cross-covariance between query and key entries. This result highlights the inherent skewness in the self-attention distribution, which is crucial for its ability to concentrate the attention and balance between exploration versus exploitation. Understanding this distributional property provides an information evidence to design the linearized attention mechanism that preserves the statistical behavior of softmax attention.

Based on this insight, the kernel-based formulation of linear attention in Equation (5) is adopted to design the feature map $\Phi(\cdot)$ to be an exponential function. In particular, this property ensures that

the resulting attention weights also follow a log-normal distribution, thereby aligning the behavior of linear attention more closely with that of softmax attention. The LLN attention was therefore derived to carry out the attended embedding as (Nahshan et al., 2023)

$$\boldsymbol{y}_t^{(\text{LLN})} = \frac{\Phi(\boldsymbol{q}_t)^\top \sum_{j=1}^N \Phi(\boldsymbol{k}_j) \boldsymbol{v}_j^\top}{\Phi(\boldsymbol{q}_t)^\top \sum_{l=1}^N \Phi(\boldsymbol{k}_l)} = \frac{(e^{\boldsymbol{q}_t})^\top \sum_{j=1}^N e^{\boldsymbol{k}_j} \boldsymbol{v}_j^\top}{(e^{\boldsymbol{q}_t})^\top \sum_{l=1}^N e^{\boldsymbol{k}_l}}. \tag{8}$$

Furthermore, the tunable hyperparameters were incorporated into the exponential feature map (Nahshan et al., 2023), allowing the distributional shape of LLN attention to precisely match that of softmax attention. In addition to this connection, LLN empirically implemented a block-diagonal attention scheme to reinforce the locality representation. These combined strategies enabled LLN attention to outperform softmax attention on multiple benchmarks, showing that linear attention is feasible to match or even exceed the performance of softmax attention when properly designed.

## 3.2 LOG-NORMAL DISTRIBUTION FOR SPARSE AND RELEVANT ATTENTION

The effectiveness of the log-normal distribution is originated from its similarity to the expected attention pattern observed in self-attention mechanism. In the self-attention as shown in Figure 3, the model tends to assign higher attention weights to a small number of relevant tokens, while distributing negligible weights to the irrelevant tokens, resulting in a sparse and concentrated attention distribution. This characteristic enables the model

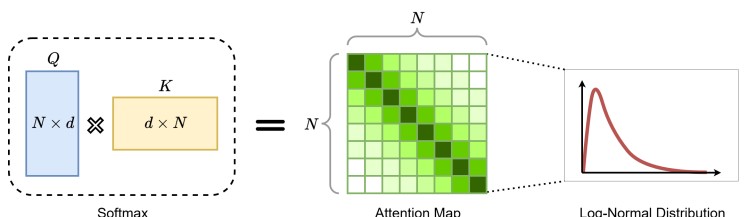

Figure 3: Illustration of the distribution for the values of self-attention. In the attention map, typically, each token strongly attends a small number of tokens (darker color) and weakly attends the other tokens (lighter color), indicating that attention primarily focuses on relatively important tokens. This pattern reflects the long-tailed shape of the attention values to be a log-normal distribution, where few values are high but most of values remain small.

to better focus on salient information within the input string, thereby improving both semantic understanding and generation accuracy. The log-normal distribution naturally exhibits this property, and it produces a long-tailed distribution in which most values lie in the low-attention region, with only a few high values corresponding to relevant tokens. This closely aligns with the desirable behavior of attention mechanism. Therefore, it is challenging to build an SSM with linear attention where the attention weights are shaped to follow a log-normal distribution. The resulting model can better reflect the behavior of self-attention and ultimately obtain improved performance.

## 4 LOG-NORMAL STATE-SPACE MODEL

### 4.1 CAUSAL MASKING

Modern generative language models (Radford et al., 2019; Touvron et al., 2023) typically adopt a decoder-only architecture, which is auto-regressive in nature. An underlying motivation of using this architecture is that, during the computation of attention map, each query at time $t$ attends only to key-value pairs from time 1 through time $t$. This causal constraint aligns with the essence of sequence generation and reduces the computational complexity compared to full self-attention.

In practice, this essence can be implemented by adding a causal mask to the attention logits. Correspondingly, after computing the dot product between query and key matrices, a lower triangular mask is applied such that the positions corresponding to future tokens are set to $-\infty$, excluding them from the softmax computation. Therefore, the masked attention (MA) is calculated by

$$\boldsymbol{y}_t^{(\text{MA})} = \frac{\sum_{j=1}^i e^{\boldsymbol{q}_t^\top \boldsymbol{k}_j} \boldsymbol{v}_j^\top}{\sum_{l=1}^i e^{\boldsymbol{q}_t^\top \boldsymbol{k}_l}}. \tag{9}$$

## 4.2 LINEAR LOG-NORMAL ATTENTION WITH CAUSAL MASKING

Although LLN (Nahshan et al., 2023), like the other linear attention mechanisms, does not explicitly compute a full attention map via query-key dot products, it can still be reformulated to incorporate causal masking. The LLN formulation with a causal summation constraint becomes:

$$
\boldsymbol{y}_t^{(\text{LLN})} = \frac{\Phi(\boldsymbol{q}_t)^\top \sum_{j=1}^{t} \Phi(\boldsymbol{k}_j)\boldsymbol{v}_j^\top}{\Phi(\boldsymbol{q}_t)^\top \sum_{l=1}^{t} \Phi(\boldsymbol{k}_l)} \tag{10}
$$

where $\Phi(\cdot)$ is an exponential function. Furthermore, we can isolate the contribution at time step $t$ from the summation to reveal the incremental structure in the calculation

$$
\boldsymbol{y}_t^{(\text{LLN})} = \frac{\Phi(\boldsymbol{q}_t)^\top \left( \sum_{j=1}^{t-1} \Phi(\boldsymbol{k}_j)\boldsymbol{v}_j^\top + \Phi(\boldsymbol{k}_t)\boldsymbol{v}_t^\top \right)}{\Phi(\boldsymbol{q}_t)^\top \left( \sum_{l=1}^{t-1} \Phi(\boldsymbol{k}_l) + \Phi(\boldsymbol{k}_t) \right)}. \tag{11}
$$

This form highlights that causal LLN attention can be naturally interpreted as a state update mechanism. The key and value contributions are accumulated over time, which mirrors the hidden state update process in recurrent models such as RNNs. Prior work (Katharopoulos et al., 2020) has also emphasized this connection between linear attention and recurrent structure.

## 4.3 STATE-SPACE MODEL WITH LOG-NORMAL DISTRIBUTION

Motivated by such a recurrent interpretation, this paper proposes the log-normal state-space model (LNSSM), which reformulates the causal LLN attention into an explicit state update form as

$$
\boldsymbol{y}_t^{(\text{LNSSM})} = \frac{\Phi(\boldsymbol{q}_t)^\top \left( S_{t-1} + \Phi(\boldsymbol{k}_t)\boldsymbol{v}_t^\top \right)}{\Phi(\boldsymbol{q}_t)^\top \left( \boldsymbol{z}_{t-1} + \Phi(\boldsymbol{k}_t) \right)} \tag{12}
$$

$$
S_t = S_{t-1} + \Phi(\boldsymbol{k}_t)\boldsymbol{v}_t^\top, \quad \boldsymbol{z}_t = \boldsymbol{z}_{t-1} + \Phi(\boldsymbol{k}_t) \tag{13}
$$

where $S_t \in \mathbb{R}^{d \times d}$ and $\boldsymbol{z}_t \in \mathbb{R}^d$ denote the accumulated key-value and key-only states, respectively. At each time step $t$, the new input $(\boldsymbol{k}_t, \boldsymbol{v}_t)$ is processed, and the feature embedding $\boldsymbol{y}_t^{(\text{LNSSM})}$ is computed by using the updated states. This formulation enables the token-by-token processing with the constant-time updates and linear memory footprint, while retaining the ability to approximate the log-normal distributional behavior of softmax attention through exponential feature maps.

## 4.4 CONNECTION BETWEEN SELF-ATTENTION AND STATE UPDATE

In general, the feature embedding of LNSSM in Equation (12) with state update in Equation (13) lacks an explicit mechanism for encoding the position information. Although causal accumulation is feasible to activate an efficient token-by-token computation, it does not contain the position information of tokens in a sequence. A naive attempt to position embedding based on the rotary position embedding (RoPE) (Su et al., 2024)) may carry out the state accumulation, potentially leading to the information smearing or numerical instability due to the unbounded numerical growth of states.

In the broader literature on SSMs, such a challenge is typically addressed through the design of a state transition matrix $A$, which governs how past the states evolve and decay over time. Recent high-performance variants of SSM, such as S4 (Gu et al., 2022), DSS (Gupta et al., 2022), S5 (Smith et al., 2022), and the advanced Mamba (Gu & Dao, 2023; Dao & Gu, 2024), all aforementioned methods emphasize the critical role of finding a carefully designed matrix $A$.

An alternative to the state decay is to utilize the forget gates (Beck et al., 2024). Gating mechanism is conceptually similar to the effect of a decay matrix. Recent studies (Hochreiter, 1997; Lin et al., 2025) have shown that forget gates were able to effectively reflect the explicit position encodings in a transformer. The underlying principle is that forget gates enable the model to learn relative position by modulating the decay dynamics of the internal state.

A well-designed matrix $A$ in SSMs is feasible to encode the temporal position through the controlled state decay. Recent work (Li et al., 2025) has further established a component-wise equivalence between SSM and transformer, showing how the attention and position encoding

in two paradigms are related. As shown in Table 1, the query $\boldsymbol{q}$ and key $\boldsymbol{k}$ in self-attention for transformer correspond naturally to the output and input matrices $C$ and $B$ in state projection and update for Mamba (Gu & Dao, 2023), respectively. LNSSM does not require the matrices $C$ and $B$ for state update since these matrices have been sufficiently reflected as the query $\boldsymbol{q}$ and key $\boldsymbol{k}$ in self-attention.

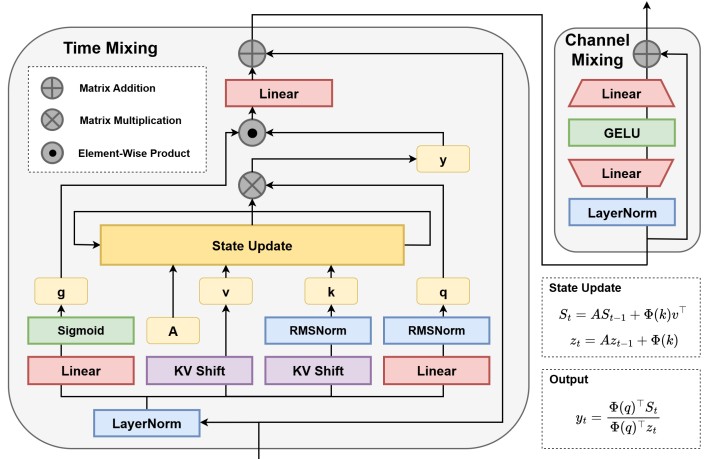

Figure 4: Time mixture and channel mixing in the architecture of LNSSM.

The state transition matrix $A$, however, plays a central role for expressiveness of a model and must be carefully designed. In various SSM variants, $A$ must be co-designed with $B$ and $C$ to properly capture the state dynamics. Given the structural similarity between LNSSM in Equations (12)-(13) and RWKV-4 (Peng et al., 2023) in Equation (2), we adopt the decay vector $w$ from RWKV-4 as our state matrix $A$, which governs the exponential decay behavior at each dimension. A full implementation of the proposed LNSSM is accordingly formulated as

$$\boldsymbol{y}_t^{(\text{LNSSM})} = \frac{(e^{\boldsymbol{q}_t})^\top \left( \sum_{j=1}^{t-1} A^{t-1-j} e^{\boldsymbol{k}_j} \boldsymbol{v}_j^\top + e^{\boldsymbol{k}_t} \boldsymbol{v}_t^\top \right)}{(e^{\boldsymbol{q}_t})^\top \left( \sum_{l=1}^{t-1} A^{t-1-l} e^{\boldsymbol{k}_l} + e^{\boldsymbol{k}_t} \right)} = \frac{\Phi(\boldsymbol{q}_t)^\top \left( AS_{t-1} + \Phi(\boldsymbol{k}_t) \boldsymbol{v}_t^\top \right)}{\Phi(\boldsymbol{q}_t)^\top \left( A\boldsymbol{z}_{t-1} + \Phi(\boldsymbol{k}_t) \right)}, \quad (14)$$

$$S_t = AS_{t-1} + \Phi(\boldsymbol{k}_t)\boldsymbol{v}_t^\top, \quad \boldsymbol{z}_t = A\boldsymbol{z}_{t-1} + \Phi(\boldsymbol{k}_t). \quad (15)$$

## 4.5 OVERALL MODEL DESIGN

To stabilize the training and mitigate the risk of exploding values due to the exponential operations, we apply the QK normalization (Dehghani et al., 2023) with RMSNorm to the feature maps $\Phi(\boldsymbol{q}_t)$ and $\Phi(\boldsymbol{k}_t)$. To enhance the local context modeling, we adopt the KV-shift (Peng et al., 2023; 2024) mechanism, which introduces a slight temporal shift to the key and value representations, helping the model capture local dependencies between

| Transformer | Mamba | LNSSM |
|---|---|---|
| $\boldsymbol{q} = W_q \boldsymbol{x}$ | $C = W_c \boldsymbol{x}$ | $\boldsymbol{q} = W_q \boldsymbol{x}$ |
| $\boldsymbol{k} = W_k \boldsymbol{x}$ | $B = W_b \boldsymbol{x}$ | $\boldsymbol{k} = W_k \boldsymbol{x}$ |
| $\boldsymbol{v} = W_v \boldsymbol{x}$ | $\boldsymbol{x} = \boldsymbol{x}$ | $\boldsymbol{v} = W_v \boldsymbol{x}$ |
| $\boldsymbol{y} = (\mathbf{pos} \circ \boldsymbol{q}^\top \boldsymbol{k}) \boldsymbol{v}^\top$ | $\boldsymbol{y} = (A \circ CB)\boldsymbol{x}$ | $\boldsymbol{y} = (A \circ \boldsymbol{q}^\top \boldsymbol{k}) \boldsymbol{v}^\top$ |

Table 1: Comparison of the transformation in transformer (Vaswani et al., 2017), Mamba (Gu & Dao, 2023) and LNSSM where $\mathbf{pos}$ denotes the position encoding and $\{W_q, W_k, W_v, W_b, W_c\}$ are the parameters.

adjacent tokens. We also introduce the output gate (Hochreiter, 1997; Beck et al., 2024) that modulates the final output based on the current state, improving the flexibility in shaping the dynamic information flow. For the MLP block, we adopt the same architecture as in GPT-2 (Radford et al., 2019), which includes a two-layer feedforward network with a GELU activation and residual connections. The complete architecture of the full LNSSM is illustrated in Figure 4.

## 5 EXPERIMENTS

### 5.1 EXPERIMENTAL SETUPS

To evaluate the effectiveness of the proposed method, this study conducted a series of experiments by augmenting the designed LNSSM block in a model architecture with alternative attention mechanisms as baselines. All baseline models shared the same setting: a 6-layer stack with

512-dimensional embeddings, resulting about 72 million parameters. Each model is trained from scratch on the SlimPajama-1B (Shen et al., 2023) dataset according to the objective for next-token prediction. During training, each epoch was run with 10K sequences of tokens with length 1024, and all models were trained for 500 epochs. The training conditions, training/test data collection, and hyperparameter settings were consistent across different models for fair comparison.

### 5.1.1 BASELINE MODELS

In the experiments, the proposed LNSSM has been shown potential and meaningful as a new type of time-mixing machine relative to the previous SSM baselines. This study compared LNSSM with the following SSM variants with different metrics.

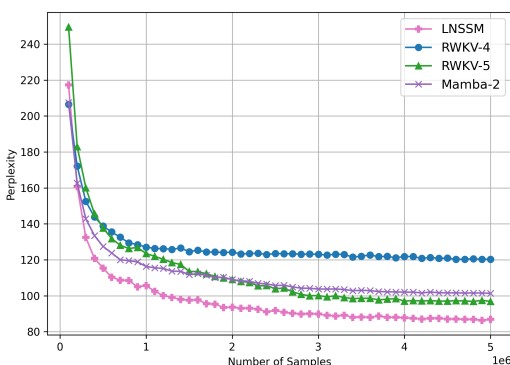

Figure 5: Training curves of the perplexity of validation data by using various time-mixing methods where the channel-mixing is consistently based on MLP. The curves are shown over the number of samples in the stochastic learning process.

**RWKV-4** We implemented the RWKV-4 (Peng et al., 2023) as a baseline model by using Equation (2). This SSM-based baseline was used in the evaluation of short-form question answering (QA) in terms of perplexity.

**RWKV-5** To evaluate the effectiveness of the log-normal distribution in attention mechanisms, we construct a baseline based on an existing SSM that does not exhibit log-normal behavior, while ensuring the architecture remains transformer-like for a fair comparison. RWKV-5 (Peng et al., 2024) satisfies these criteria, as its formulation in Equation (3) resembles that of a transformer and can serve as a suitable benchmark. We use RWKV-5 as the SSM baseline without log-normal distribution.

**Mamba-2** Mamba-2 Dao & Gu (2024) is a recently proposed state-space model that builds upon its predecessor, Mamba Gu & Dao (2023), for further improvements. Specifically, Mamba-2 simplifies the time-varying state matrix $A$ in Mamba by replacing it with a time-invariant style in

$$\boldsymbol{h}_t = A\boldsymbol{h}_{t-1} + B_t\boldsymbol{x}_t, \quad \boldsymbol{y}_t = C_t\boldsymbol{h}_t \tag{16}$$

while still maintaining the competitive performance.

**Transformer** We also implemented a standard transformer driven by RoPE (Su et al., 2024) position encoding and full self-attention. This baseline was evaluated on long-form QA, as well as for computational cost and memory usage analysis.

### 5.2 PERPLEXITY EVALUATION

This study conducted the evaluation of language modeling for next token prediction based on different methods by using the validation set of SlimPajama-1B. As shown in Figure 5, LNSSM outperform RWKV-4 across the training epochs. Moreover, compared to the non-log-normal SSM baseline, e.g. RWKV-5, LNSSM achieves significantly lower perplexity, indicating the benefit of incorporating a log-normal distribution in attention modeling.

### 5.3 CONTEXT TRACKING EVALUATION

| Model | qa1 | qa2 | qa3 | qa4 | qa5 | qa6 | qa7 | qa8 | qa9 | qa10 | qa11 | qa12 | qa13 | qa14 | qa15 | qa16 | qa17 | qa18 | qa19 | qa20 |
|---|---|---|---|---|---|---|---|---|---|---|---|---|---|---|---|---|---|---|---|---|
| Mamba-2 | 48.31 | **36.17** | 25.9 | 54.41 | **56.52** | 57.84 | 55.79 | 42.63 | 57.25 | **51.12** | 68.51 | 58.73 | 86.26 | 29.58 | 48.74 | 40.03 | 51.47 | 65.56 | 0.00 | **54.17** |
| RWKV-4 | 45.95 | 32.10 | 24.31 | 49.54 | 50.97 | 52.41 | 55.38 | 43.18 | 58.36 | 50.77 | 67.28 | 56.31 | 76.72 | 26.05 | 49.03 | 41.44 | 52.62 | **72.31** | 0.00 | 53.23 |
| RWKV-5 | 48.62 | 35.38 | **29.85** | **54.67** | 56.10 | 53.03 | 57.44 | **45.03** | 55.79 | 45.03 | 70.46 | 56.21 | 82.97 | **37.44** | 49.13 | 38.97 | 53.54 | 60.10 | 0.00 | 38.36 |
| LNSSM | **50.15** | 32.00 | 21.54 | 51.90 | 49.64 | **63.18** | **57.74** | 44.72 | **64.72** | 48.92 | **71.38** | **64.31** | **88.00** | 24.41 | **53.13** | **41.95** | **54.87** | 71.18 | 0.00 | 47.90 |

Table 2: Accuracy (%) of different models on BABILong 0k QA1–20. The best accuracy per QA is in **bold**.

To further evaluate the reasoning and contextual tracking abilities of each model, we adopt the BABILong (Kuratov et al., 2024) benchmark, which consists of question-answering tasks across different input sequence lengths including 0k, 1k, 2k, 4k and 8k tokens. The 0k setting corresponds to short-form QA, while the other configurations are used for long-form QA evaluation.

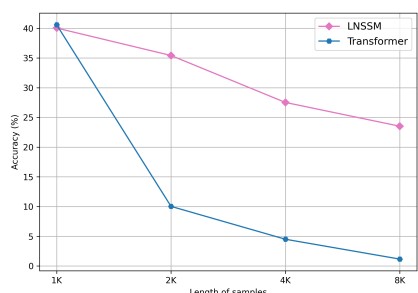

Due to the relatively small model size, all models exhibit limited expressiveness in zero-shot settings. Therefore, we employ few-shot fine-tuning to adapt the models before evaluation. For short-form QA (0k), we fine-tune each model by using 5% of the 0k training data over 100 epochs, reusing the same samples in each epoch. For long-form QA, we fine-tune by using 10% of the 1k training set, and evaluate on mixed data from the 2k, 4k and

Figure 6: Accuracy (%) of transformer with self-attention and LNSSM with state update where the BABILong 1k to 8k is evaluated.

8k tasks. Table 2 summarizes the results on short-form QA. LNSSM achieves the highest accuracy on most tasks. In contrast, RWKV-5, the non-log-normal SSM only performs competitively on a few tasks and fails to generalize well. Figure 6 illustrates the performance on long-form QA tasks. As expected, the transformer with self-attention and RoPE encoding suffers from significant performance degradation as sequence length increases. This highlights the limitations of self-attention in transformer in capturing long-range dependencies. In contrast, LNSSM maintains robust performance across the increasing sequence lengths, with only mild degradation due to the increased task complexity rather than the memory limitation.

## 5.4 LONG-FORM GENERATION EVALUATION

We evaluated LNSSM against the other SSM variants on a set of long-form generation tasks. Specifically, the ASQA (Stelmakh et al., 2022) dataset was used to assess the model's ability to locate and comprehensively express answers within a given long textual description. The ELI5 (Fan et al., 2019) dataset was employed to evaluate the model's knowledge retention and generation capabilities in the absence of reference material. Finally, the WikiLarge (Zhang & Lapata, 2017) dataset was utilized to examine the model's ability to read an entire long-form document and subsequently summarize or paraphrase it. For all evaluations, ROUGE-L was adopted

|  | Zero-shot | | | Fine-tuned | |
|---|---|---|---|---|---|
| Model | ASQA | ELI5 | WikiLarge | ASQA | WikiLarge |
| Mamba-2 | 13.25 | 11.43 | 9.03 | 20.49 | 21.97 |
| RWKV-4 | 13.36 | 11.07 | 7.16 | 14.24 | 17.57 |
| RWKV-5 | 12.41 | **12.01** | 8.6 | 21.29 | 21.26 |
| LNSSM | **13.69** | 11.96 | **9.51** | **21.45** | **22.13** |

Table 3: ROUGE-L score of different models on ASQA, ELI5 and WikiLarge. The highest score per dataset is in **bold**.

as the primary evaluation metric. The models were first assessed in a zero-shot setting using the test sets of ASQA, ELI5 and WikiLarge. The results, presented in Table 3, indicate that although the relatively small parameter size of the evaluated models limits absolute performance, LNSSM consistently outperforms different SSM variants in terms of generation quality. Subsequently, we fine-tuned the models on the training sets of ASQA and WikiLarge. Detailed fine-tuning configurations are provided in Appendix B.2. As shown in Table 3, all models exhibit substantial performance improvements after fine-tuning, with LNSSM achieving superior generation results compared to Mamba-2, RWKV-4 and RWKV-5.

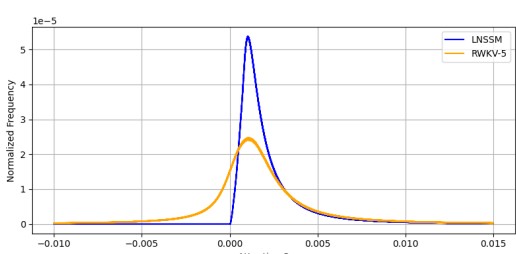

Figure 7: Distributions of the attention scores calculated by SSM via RWKV-5 and LNSSM.

## 5.5 DISTRIBUTION ANALYSIS

Although LNSSM does not explicitly compute a query-key attention map during the forward pass, we perform an auxiliary analysis to examine the distributional property of its attention scores. Specifically, we compute the dot product between the query and key representations to form an $N \times N$ synthetic attention map, and then collect all scalar values in the map to plot their empirical distribution. We compare the attention score distributions of the non-log-normal SSM variant via RWKV-5 and the LNSSM. As shown in Figures 7 (overall) and 8 (per sequence), the results align with our expectations. The attention scores of the standard SSM via RWKV-5 are approximately normal distributed while the atten-

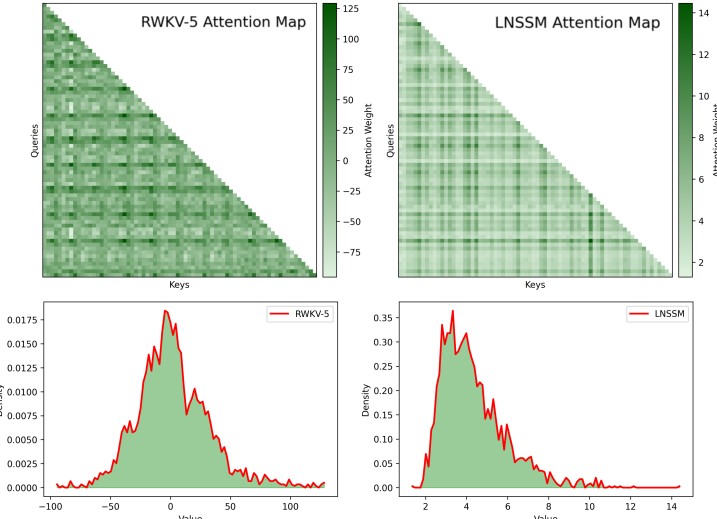

Figure 8: Comparison of attention maps without and without log-normal distribution. For a fair comparison, we use the *unnormalized* attention maps. the attention map of RWKV-5 (which does not exhibit a log-normal distribution) shows smaller differences across tokens. In contrast, the attention map of LNSSM, which follows a log-normal distribution, better distinguishes between relevant and irrelevant tokens. In terms of distribution, RWKV-5 shows a normal distribution while LNSSM shows a log-normal distribution.

tion scores of LNSSM follow a log-normal distribution. In attention mechanisms, it is typically desirable for the model to focus selectively on a small subset of important or relevant tokens. That is, the attention weights should be highly skewed, with a few large values corresponding to salient tokens and the rest being small.

The log-normal distribution naturally exhibits this behavior, it is positively skewed, with the majority of values concentrated near zero and a long tail of high values. This property helps explain why the causal state update based on LNSSM even achieves the performance comparable to the noncausal self-attention. A desirable performance basically relies on sparse and relevant attention distributions. In contrast, a normal distribution, as observed in the non-log-normal SSM (RWKV-5), implies that most attention scores are centered around the mean, with fewer extreme values. This results in a less focused attention pattern, where many tokens receive similar weights, deviating from the ideal behavior expected of effective attention. These findings highlight that distributional alignment based on softmax attention with log-normality is a key factor in achieving strong performance, and support the design choice of using exponential mappings in LNSSM to emulate this property.

## 6 CONCLUSIONS

This paper has presented the tog-normal state-space model (LNSSM), which inherits the statistical property of self-attention while enjoying the efficiency of state space models. Experiments show that LNSSM matches the expressiveness of self-attention in transformer on the benchmarks with different training-length sequences, while outperforming standard SSM baselines that lack log-normality.

For long-range dependencies, LNSSM benefits from its state matrix memory, maintaining stable performance on the sequence data with extended lengths where transformer considerably degrades, likely due to overfitting to fixed training lengths. On efficiency, LNSSM achieves true linear complexity in floating point operations and memory, requiring only constant memory as it processes tokens sequentially. In contrast, self-attention in transformer scales quadratically or worse in computation, and KV caching, while reducing the computation, still incurs memory overhead. Overall, LNSSM offers a strong balance between expressiveness and efficiency, making it a promising alternative to attention-based architectures under the memory and computation constraints.

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

APPENDIX

## A  HARDWARE-EFFICIENT IMPLEMENTATION

Transformer-based models rely on the self-attention mechanism during inference, which requires computing attention scores between every pair of tokens in a sequence of length $N$. Consequently, the time complexity per token is $\mathcal{O}(N^2)$. Moreover, since generating the $N$-th token depends on the full context of tokens 1 through $N-1$, the overall inference complexity sums to $1^2 + 2^2 + \cdots + (N-1)^2 = \mathcal{O}(N^3)$. However, during training, the entire sequence of length $N$ is known in advance, enabling the simultaneous input of all tokens and parallel prediction of the next tokens at each position. This design allows efficient parallelization by processing all tokens concurrently.

SSMs can also adopt a training scheme that feeds in $N$ tokens at once. Nonetheless, they do not support full parallelization in the same manner. This is because the generation of each token depends not only on the current input but also on the previous state. In particular, producing the $N$-th token requires the state from the $(N-1)$-th step. Therefore, token generation in SSMs must proceed sequentially, preventing simultaneous processing of all $N$ tokens in a true parallel fashion.

This characteristic significantly impacts training efficiency. In practice, transformer-based models process sequences of length $N$ by batching all $N$ token embeddings together and feeding them into the GPU for parallel computation, minimizing CPU-GPU data transfers and maximizing hardware utilization. In contrast, SSMs rely on sequential recurrence to process inputs token by token. When implemented in standard deep learning frameworks such as PyTorch, each token's processing may involve frequent synchronization between CPU and GPU. For sequences of length $N$, this synchronization occurs $N$ times, causing the GPU to frequently wait for data from the CPU. This overhead results in underutilized GPU resources and reduced training throughput.

Although existing research on state-space models (SSMs) has proposed optimization strategies such as "space-for-time" trade-offs (e.g., storing partial intermediate states to support parallelization), these methods primarily focus on improving theoretical computational complexity without fundamentally addressing the practical issue of low GPU utilization. Therefore, enhancing GPU resource usage efficiency remains a critical challenge for accelerating training.

One promising approach to overcome this bottleneck is the implementation of custom CUDA kernels. The core idea is to batch the entire input sequence and transfer it once to the GPU, where the sequential recurrence logic is executed entirely within the CUDA kernel. This design enables all token processing to be performed internally on the GPU, and only the final results are transferred back to the CPU after processing the whole sequence. This approach significantly reduces host-device synchronization overhead and markedly improves GPU utilization. Table 4 illustrates the difference in GPU usage and training time between implementations using custom CUDA kernels and those relying solely on PyTorch under identical conditions.

| Method | GPU usage | Training time |
|---|---|---|
| **Pytorch** | 58% | 4260 |
| **CUDA kernel** | 99% | 97 |

Table 4: GPU usage in percentage and training time in seconds per epoch when using LNSSM.

CUDA kernels parallelize the calculations for each dimension of a vector. In other words, if we need to calculate a vector of $d$ dimensions, the CUDA kernel can simultaneously compute the results for all $d$ dimensions, achieving a high degree of parallelism. However, extreme caution is required when using the shared variables, as improper handling can lead to race conditions. Furthermore, implementing custom CUDA kernels poses significant challenges, as PyTorch's automatic differentiation cannot be directly utilized within CUDA kernels. Therefore, all computations inside the kernel must be manually coded along with their gradient calculations. This is a substantial engineering effort, especially because the SSM state accumulates over time. The state computed at timestep $t$ is used not only to generate the output at time $t$, but also retained and accumulated to produce the state and output at timestep $t+1$. In other words, the loss at timestep $t$ requires gradient calculations not only for the parameters at $t$, but also for all parameters from timestep 1 through $t-1$. Below, we present the gradient formulas of each parameter in LNSSM along with the corresponding gradient computation algorithms. To better leverage PyTorch's native automatic differentiation, we only practically compute the denominator $S$ and the numerator $z$ of the LNSSM equations within the CUDA kernel,

thereby simplifying the gradient calculations inside the CUDA kernel. The following algorithms describe how to compute the gradients of the denominator $S$ with respect to each parameter.

The gradient of the loss with respect to $q$ is given as follows

$$\frac{\partial \mathcal{L}}{\partial \boldsymbol{q}_n^t} = \sum_{i=1}^{n} \frac{\partial \mathcal{L}}{\partial \boldsymbol{y}_i^t} \cdot \frac{\partial \boldsymbol{y}_i^t}{\partial \boldsymbol{q}_n^t} = \frac{\partial \mathcal{L}}{\partial \boldsymbol{y}_1^t} S_{n1}^t + \frac{\partial \mathcal{L}}{\partial \boldsymbol{y}_2^t} S_{n2}^t + \cdots + \frac{\partial \mathcal{L}}{\partial \boldsymbol{y}_n^t} S_{nn}^t \qquad (17)$$

where $\boldsymbol{y}$ denotes the output produced by the CUDA kernel, which is then received by the external Python program. During backpropagation, the CUDA kernel obtains the gradient of the loss with respect to $\boldsymbol{y}$ from the external Python program and uses it to compute the parameter gradients based on chain rule, as described in Algorithm 1.

---

**Algorithm 1** Per-dimension kernel for $\frac{\partial S}{\partial \boldsymbol{q}_n^t}$ (thread computes a fixed $n$)

---

**Require:**
1: $T$ **:** number of time steps (indexed $t = 0, 1, \ldots, T - 1$)
2: $N$ **:** feature dimension (indexed $i, n = 0, 1, \ldots, N - 1$)
3: $\boldsymbol{k}[0 \ldots T - 1][0 \ldots N - 1]$ **:** keys
4: $\boldsymbol{v}[0 \ldots T - 1][0 \ldots N - 1]$ **:** values
5: $\boldsymbol{g}\_S[0 \ldots T - 1][0 \ldots N - 1]$ **:** upstream gradient w.r.t. $S$ (from outer backprop)
6: $\boldsymbol{a}[0 \ldots N - 1]$ **:** time-invariant coefficients (per-dimension)
7: $n$ **:** the target dimension for this kernel instance (thread-local)
8: **Init:** state$[0 \ldots N - 1] \leftarrow 0$               (private to thread $n$)
**Ensure:**
9: $\boldsymbol{g}\_q[0 \ldots T - 1]$ for this $n$ **:** gradient w.r.t. $\boldsymbol{q}_n^t$ across all $t$
10: **(Collecting over all threads $n$ yields $\boldsymbol{g}\_q[0 \ldots T - 1][0 \ldots N - 1]$)**
11: **for** $t = 0$ **to** $T - 1$ **do**
12:     $\boldsymbol{g}\_q[t] \leftarrow 0$;
13:     **for** $i = 0$ **to** $N - 1$ **do**
14:         $\boldsymbol{s} \leftarrow$ state$[i]$;
15:         $\boldsymbol{x} \leftarrow \boldsymbol{k}[t][n] \cdot \boldsymbol{v}[t][i]$;
16:         $\boldsymbol{s} \leftarrow \boldsymbol{s} \cdot \boldsymbol{a}[n] + \boldsymbol{x}$;
17:         state$[i] \leftarrow \boldsymbol{s}$;
18:         $\boldsymbol{g}\_q[t] \leftarrow \boldsymbol{g}\_q[t] + \boldsymbol{s} \cdot \boldsymbol{g}\_S[t][i]$;
19:     **end for**
20: **end for**
21: **return** $\boldsymbol{g}\_q[0 \ldots T - 1]$

---

The gradient of the loss with respect to $\boldsymbol{a}$ is formulated as follows

$$\frac{\partial \mathcal{L}}{\partial \boldsymbol{a}_n} = \sum_{i=1}^{n} \frac{\partial \mathcal{L}}{\partial \boldsymbol{y}_i^t} \cdot \frac{\partial \boldsymbol{y}_i^t}{\partial \boldsymbol{a}_n} = \boldsymbol{q}_n^t \left( \frac{\partial \mathcal{L}}{\partial \boldsymbol{y}_1^t} \cdot \frac{\partial S_{n1}^t}{\partial \boldsymbol{a}_n} + \frac{\partial \mathcal{L}}{\partial \boldsymbol{y}_2^t} \cdot \frac{\partial S_{n2}^t}{\partial \boldsymbol{a}_n} + \cdots + \frac{\partial \mathcal{L}}{\partial \boldsymbol{y}_n^t} \cdot \frac{\partial S_{nn}^t}{\partial \boldsymbol{a}_n} \right). \qquad (18)$$

Since $\boldsymbol{a}$ is time-invariant, it is necessary to additionally account for the derivative of the state $S$ with respect to $\boldsymbol{a}$, which is expressed as

$$\frac{\partial S_{nm}^T}{\partial \boldsymbol{a}_n} = (T - 1)(\boldsymbol{a}_n)^{T-2} \boldsymbol{k}_n^1 \boldsymbol{v}_m^1 + (T - 2)(\boldsymbol{a}_n)^{T-3} \boldsymbol{k}_n^2 \boldsymbol{v}_m^2 + \cdots + \boldsymbol{k}_n^{T-1} \boldsymbol{v}_m^{T-1}. \qquad (19)$$

The corresponding computation procedure is presented in Algorithm 2.

---

**Algorithm 2** Per-dimension kernel for $\frac{\partial S}{\partial \boldsymbol{a}_n}$ (thread computes a fixed $n$)

---

**Require:**
1: $T$ **:** number of time steps (indexed $t = 0, 1, \ldots, T - 1$)
2: $N$ **:** feature dimension (indexed $i, n = 0, 1, \ldots, N - 1$)
3: $\boldsymbol{k}[0 \ldots T - 1][0 \ldots N - 1]$ **:** keys
4: $\boldsymbol{v}[0 \ldots T - 1][0 \ldots N - 1]$ **:** values
5: $\boldsymbol{q}[0 \ldots T - 1][0 \ldots N - 1]$ **:** query parameters (time-varying)
6: $\boldsymbol{g}\_S[0 \ldots T - 1][0 \ldots N - 1]$ **:** upstream gradient from outer backprop
7: $\boldsymbol{a}[0 \ldots N - 1]$ **:** time-invariant coefficients
8: $n$ **:** target dimension (thread-local index)
9: **Init:** state_1$[0 \ldots N - 1] \leftarrow 0$, state_2$[0 \ldots N - 1] \leftarrow 0$
**Ensure:**
10: $\boldsymbol{g}\_a[n]$ **:** scalar gradient w.r.t. $\boldsymbol{a}_n$ (per dimension)
11: $\boldsymbol{g}\_a \leftarrow 0$;
12: **for** $t = 0$ **to** $T - 1$ **do**
13:    $tmp\_1 \leftarrow 0$;
14:    **for** $i = 0$ **to** $N - 1$ **do**
15:       $\boldsymbol{s}\_1 \leftarrow$ state_1$[i]$;
16:       $\boldsymbol{s}\_2 \leftarrow$ state_2$[i]$;
17:       $\boldsymbol{x} \leftarrow \boldsymbol{k}[t][n] \cdot \boldsymbol{v}[t][i]$;
18:       $tmp\_2 \leftarrow \boldsymbol{a}[n] \cdot (\boldsymbol{x} + \boldsymbol{s}\_1)$;
19:       $\boldsymbol{s}\_1 \leftarrow tmp\_2$;
20:       $\boldsymbol{s}\_2 \leftarrow tmp\_2 + \boldsymbol{a}[n] \cdot \boldsymbol{s}\_2$;
21:       $tmp\_1 \leftarrow tmp\_1 + \boldsymbol{s}\_2 \cdot \boldsymbol{g}\_S[t][i]$;
22:       state_1$[i] \leftarrow \boldsymbol{s}\_1$;
23:       state_2$[i] \leftarrow \boldsymbol{s}\_2$;
24:    **end for**
25:    $\boldsymbol{g}\_a \leftarrow \boldsymbol{g}\_a + \boldsymbol{q}[t][n] \cdot tmp\_1$;
26: **end for**
27: **return** $\boldsymbol{g}\_a$ (store into $g\_a[n]$)

---

The gradient of the loss with respect to $k$ is given as follows

$$\frac{\partial \mathcal{L}}{\partial \boldsymbol{k}_n^t} = \sum_{i=1}^{n} \frac{\partial \mathcal{L}}{\partial \boldsymbol{y}_i^t} \frac{\partial \boldsymbol{y}_i^t}{\partial \boldsymbol{k}_n^t} + K_n^{t+1 \sim T} = \boldsymbol{q}_n^t \left( \frac{\partial \mathcal{L}}{\partial \boldsymbol{y}_1^t} \boldsymbol{v}_1^t + \frac{\partial \mathcal{L}}{\partial \boldsymbol{y}_2^t} \boldsymbol{v}_2^t + \cdots + \frac{\partial \mathcal{L}}{\partial \boldsymbol{y}_n^t} \boldsymbol{v}_n^t \right) + K_n^{t+1 \sim T} \quad (20)$$

where $K$ is the future impact of $\boldsymbol{k}$. The gradient of the loss with respect to $\boldsymbol{v}$ is given as follows

$$\frac{\partial \mathcal{L}}{\partial \boldsymbol{v}_n^t} = \frac{\partial \mathcal{L}}{\partial \boldsymbol{y}_n^t} \frac{\partial \boldsymbol{y}_n^t}{\partial \boldsymbol{v}_n^t} + V_n^{t+1 \sim T} = \frac{\partial \mathcal{L}}{\partial \boldsymbol{y}_n^t} \left( \boldsymbol{q}_1^t \boldsymbol{k}_1^t + \boldsymbol{q}_2^t \boldsymbol{k}_2^t + \cdots + \boldsymbol{q}_n^t \boldsymbol{k}_n^t \right) + V_n^{t+1 \sim T} \quad (21)$$

where $V$ is the future impact of $\boldsymbol{v}$. Since the state is derived from the multiplication of $\boldsymbol{k}$ and $\boldsymbol{v}$, the current values of $\boldsymbol{k}$ and $\boldsymbol{v}$ also influence future outputs. Therefore, when computing the gradients of $\boldsymbol{k}$ and $\boldsymbol{v}$, their future impacts must be considered. The future impact of $\boldsymbol{k}$ is computed as follows

$$\begin{aligned}
K_n^{t+1 \sim T} = \boldsymbol{q}_n^{t+1} \boldsymbol{a}_n &\left( \frac{\partial \mathcal{L}}{\partial \boldsymbol{y}_1^{t+1}} \boldsymbol{v}_1^t + \frac{\partial \mathcal{L}}{\partial \boldsymbol{y}_2^{t+1}} \boldsymbol{v}_2^t + \cdots + \frac{\partial \mathcal{L}}{\partial \boldsymbol{y}_n^{t+1}} \boldsymbol{v}_n^t \right) \\
&+ \boldsymbol{q}_n^{t+2} (\boldsymbol{a}_n)^2 \left( \frac{\partial \mathcal{L}}{\partial \boldsymbol{y}_1^{t+2}} \boldsymbol{v}_1^t + \frac{\partial \mathcal{L}}{\partial \boldsymbol{y}_2^{t+2}} \boldsymbol{v}_2^t + \cdots + \frac{\partial \mathcal{L}}{\partial \boldsymbol{y}_n^{t+2}} \boldsymbol{v}_n^t \right) \\
&+ \cdots \\
&+ \boldsymbol{q}_n^T (\boldsymbol{a}_n)^{T-t} \left( \frac{\partial \mathcal{L}}{\partial \boldsymbol{y}_1^T} \boldsymbol{v}_1^t + \frac{\partial \mathcal{L}}{\partial \boldsymbol{y}_2^T} \boldsymbol{v}_2^t + \cdots + \frac{\partial \mathcal{L}}{\partial \boldsymbol{y}_n^T} \boldsymbol{v}_n^t \right).
\end{aligned} \quad (22)$$

---

**Algorithm 3** Per-dimension kernel for $\frac{\partial S}{\partial \boldsymbol{k}_n^t}$ (thread computes a fixed $n$)

---

**Require:**
 1: $T$ **:** number of time steps (indexed $t = 0, 1, \ldots, T - 1$)
 2: $N$ **:** feature dimension (indexed $i, n = 0, 1, \ldots, N - 1$)
 3: $\boldsymbol{q}[0 \ldots T - 1][0 \ldots N - 1]$ **:** query parameters (time-varying)
 4: $\boldsymbol{v}[0 \ldots T - 1][0 \ldots N - 1]$ **:** values
 5: $\boldsymbol{g}\_S[0 \ldots T - 1][0 \ldots N - 1]$ **:** upstream gradient from outer backprop
 6: $\boldsymbol{a}[0 \ldots N - 1]$ **:** time-invariant coefficients
 7: $n$ **:** target dimension (thread-local index)
 8: **Init:** `state_3`$[0 \ldots N - 1] \leftarrow 0$
**Ensure:**
 9: $\boldsymbol{g}\_k[0 \ldots T - 1]$ for this $n$ **:** gradient w.r.t. $\boldsymbol{k}_n^t$ across all $t$
10: **(Collecting over all threads $n$ yields $\boldsymbol{g}\_k[0 \ldots T - 1][0 \ldots N - 1]$)**
11: **for** $t = T - 1$ **to** $0$ **step** $-1$ **do**
12:     $g\_k[t] \leftarrow 0$;
13:     **for** $i = 0$ **to** $N - 1$ **do**
14:         $\boldsymbol{s} \leftarrow$ `state_3`$[i]$;
15:         $\boldsymbol{x} \leftarrow \boldsymbol{q}[t][n] \cdot \boldsymbol{g}\_S[t][i]$;
16:         $\boldsymbol{s} \leftarrow \boldsymbol{x} + \boldsymbol{s} \cdot \boldsymbol{a}[n]$;
17:         `state_3`$[i] \leftarrow \boldsymbol{s}$;
18:         $\boldsymbol{g}\_k[t] \leftarrow \boldsymbol{g}\_k[t] + \boldsymbol{s} \cdot \boldsymbol{v}[t][i]$;
19:     **end for**
20: **end for**
21: **return** $\boldsymbol{g}\_k[0 \ldots T - 1]$

---

The future impact of $\boldsymbol{v}$ is computed as follows

$$
\begin{aligned}
V_n^{t+1 \sim T} = & \frac{\partial \mathcal{L}}{\partial \boldsymbol{y}_n^{t+1}} \Big( \boldsymbol{q}_1^{t+1} \boldsymbol{a}_1 \boldsymbol{k}_1^t + \boldsymbol{q}_2^{t+1} \boldsymbol{a}_2 \boldsymbol{k}_2^t + \cdots + \boldsymbol{q}_n^{t+1} \boldsymbol{a}_n \boldsymbol{k}_n^t \Big) \\
& + \frac{\partial \mathcal{L}}{\partial \boldsymbol{y}_n^{t+2}} \Big( \boldsymbol{q}_1^{t+2} (\boldsymbol{a}_1)^2 \boldsymbol{k}_1^t + \boldsymbol{q}_2^{t+2} (\boldsymbol{a}_2)^2 \boldsymbol{k}_2^t + \cdots + \boldsymbol{q}_n^{t+2} (\boldsymbol{a}_n)^2 \boldsymbol{k}_n^t \Big) \\
& + \cdots \\
& + \frac{\partial \mathcal{L}}{\partial \boldsymbol{y}_n^T} \Big( \boldsymbol{q}_1^T (\boldsymbol{a}_1)^{T-t} \boldsymbol{k}_1^t + \boldsymbol{q}_2^T (\boldsymbol{a}_2)^{T-t} \boldsymbol{k}_2^t + \cdots + \boldsymbol{q}_n^T (\boldsymbol{a}_n)^{T-t} \boldsymbol{k}_n^t \Big).
\end{aligned}
\tag{23}
$$

The detailed procedures are described in Algorithms 3 and 4, respectively.

## B  EXPERIMENTAL DETAILS

### B.1  TRAINING

For a smaller dataset, each sample was concatenated to form a continuous long text, which served as the training corpus. We employed the same byte pair encoding (BPE) tokenizer as GPT-2, which tokenizes frequent words in the corpus as single tokens, adding a special leading symbol to indicate preceding whitespace. Rare words were decomposed into subword units. The total vocabulary size was 50,277.

To reduce the training time and hardware requirement, a larger dataset was preprocessed into contiguous binary files (.bin) containing token ID sequences, accompanied by index files (.idx) storing the byte offsets of each sequence. This preprocessing allows for efficient sequential reading and random access during training, eliminating the need for on-the-fly text parsing and tokenization.

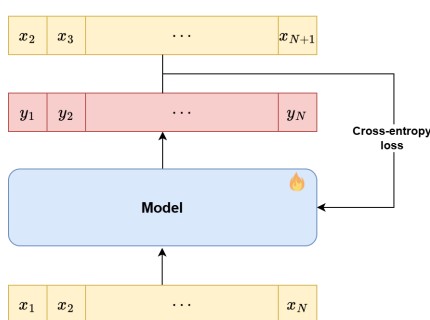

Figure 9: Training tasks for all models.

Such a format minimizes the disk input/output overhead, enables the memory-mapped loading, and

---

**Algorithm 4** Per-dimension kernel for $\frac{\partial S}{\partial \boldsymbol{v}_n^t}$ (thread computes a fixed $n$)

---

**Require:**
1: $T$ **:** number of time steps (indexed $t = 0, 1, \ldots, T - 1$)
2: $N$ **:** feature dimension (indexed $i, n = 0, 1, \ldots, N - 1$)
3: $\boldsymbol{q}[0 \ldots T - 1][0 \ldots N - 1]$ **:** query parameters (time-varying)
4: $\boldsymbol{k}[0 \ldots T - 1][0 \ldots N - 1]$ **:** keys
5: $\boldsymbol{g}\_S[0 \ldots T - 1][0 \ldots N - 1]$ **:** upstream gradient from outer backprop
6: $\boldsymbol{a}[0 \ldots N - 1]$ **:** time-invariant coefficients
7: $n$ **:** target dimension (thread-local index)
8: **Init:** $\texttt{state\_4}[0 \ldots N - 1] \leftarrow 0$
**Ensure:**
9: $\boldsymbol{g}\_v[0 \ldots T - 1]$ for this $n$ **:** gradient w.r.t. $\boldsymbol{v}_n^t$ across all $t$
10: **(Collecting over all threads $n$ yields $\boldsymbol{g}\_v[0 \ldots T - 1][0 \ldots N - 1]$)**
11: **for** $t = T - 1$ **to** $0$ **step** $-1$ **do**
12: $\quad g\_v[t] \leftarrow 0$;
13: $\quad$ **for** $i = 0$ **to** $N - 1$ **do**
14: $\quad\quad \boldsymbol{s} \leftarrow \texttt{state\_4}[i]$;
15: $\quad\quad \boldsymbol{x} \leftarrow \boldsymbol{g}\_S[t][n] \cdot \boldsymbol{q}[t][i]$;
16: $\quad\quad \boldsymbol{s} \leftarrow \boldsymbol{x} + \boldsymbol{s} \cdot \boldsymbol{a}[i]$;
17: $\quad\quad \texttt{state\_4}[i] \leftarrow \boldsymbol{s}$;
18: $\quad\quad \boldsymbol{g}\_v[t] \leftarrow \boldsymbol{g}\_v[t] + \boldsymbol{s} \cdot \boldsymbol{k}[t][i]$;
19: $\quad$ **end for**
20: **end for**
21: **return** $\boldsymbol{g}\_v[0 \ldots T - 1]$

---

ensures that the training pipeline can directly sample the fixed-length segments without re-tokenizing the dataset, thereby improving throughput and scalability.

### B.2 FINE-TUNING

In each epoch, a fixed number of sequences were randomly sampled from the corpus, each with a sequence length of 1024 tokens. Samples were batched according to the batch size and fed into the model for training. Validation was performed every ten epochs, during which the full validation set was processed once. For the models with approximately 70M parameters, we sampled 10,000 fixed-length sequences per epoch with a batch size of 18.

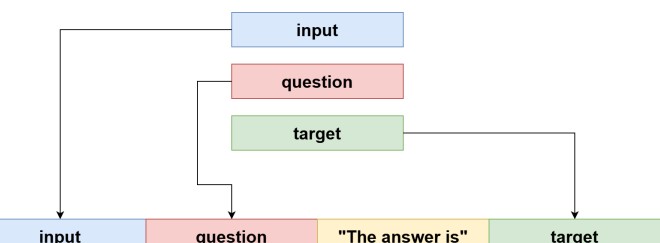

Figure 10: Data pre-processing of the BABILong dataset.

The training task was known as the next-token prediction. To maximize the efficiency, the model was trained to predict the entire sequence in parallel. For each sequence of length $N + 1$, the data sample was split into an input segment (tokens 1 to $N$) and a target segment (tokens 2 to $N + 1$). The model received a token $i$ as input and, using its updated hidden state, to predict token $i + 1$. This process is illustrated in Figure 9. For a sequence of length $N + 1$, $\boldsymbol{x}_1$ to $\boldsymbol{x}_{N+1}$, the model computes the cross-entropy loss between the predictions $\boldsymbol{y}_1$ to $\boldsymbol{y}_N$ and the target tokens $\boldsymbol{x}_2$ to $\boldsymbol{x}_{N+1}$.

The BABILong dataset consists of three fields: input, question, and target. The input is a long-form textual description containing the states of various objects. The question queries the final state of a specific object, and the target is typically a single token. For training and evaluation convenience, these three fields were concatenated sequentially, with the phrase "The answer is" inserted between the question and the target. The target token was then appended to complete the sequence, as shown in Figure 10. This formulation allows the fine-tuning process to remain consistent with the next-token prediction framework, i.e. the model is fed with the concatenated text (input + question + prompt) and trained to predict the next token, which corresponds to the answer.

# C    RESOURCE USAGE

We evaluate the computational and memory efficiency of state update in LNSSM in comparison with self-attention in transformer, both with and without KV caching. Specifically, we measure the floating point operations (FLOPs) and peak memory usage required to autoregressively generate 256, 512, 1024 tokens.

| Model | LNSSM | SA | SA with KV-cache |
|---|---|---|---|
| Time Complexity | $\mathcal{O}(Nd^2)$ | $\mathcal{O}(N^3d)$ | $\mathcal{O}(N^2d)$ |
| 256 tokens | 0.37 | 291.25 | 1.41 |
| 512 tokens | 0.74 | 1312.16 | 3.22 |
| 1024 tokens | 1.48 | 6329.43 | 8.06 |

Table 5: GFLOPs of LNSSM with state update, self-attention (SA) and SA with key-value (KV) cache to generate 256, 512, and 1024 tokens.

It is important to note that the vanilla self-attention in transformer requires the entire input sequence at each time step, resulting in a theoretical $\mathcal{O}(N^3)$ time complexity for generating $N$ tokens. In contrast, transformer with KV caching reduces this complexity to $\mathcal{O}(N^2)$ by storing and reusing previously computed key-value pairs. LNSSM, by design, supports token-by-token computation with constant state updates, yielding an expected linear $\mathcal{O}(N)$ scaling.

Table 5 presents the measured FLOPs for each model across different generation lengths. As expected, LNSSM exhibits linear growth in FLOPs with respect to output length. The self-attention transformer and its KV-cached variant deviate slightly from their theoretical scaling due to the presence of linear matrix operations (e.g., MLPs, projections) whose cost increases with input length, particularly in practical implementations.

| Model | LNSSM | SA | SA with KV-cache |
|---|---|---|---|
| 256 tokens | 0.5 GB | 1.0 GB | 1.7 GB |
| 512 tokens | 0.5 GB | 1.8 GB | 3.8 GB |
| 1024 tokens | 0.5 GB | 5.0 GB | 13.2 GB |

Table 6: Memory usage of LNSSM with state update, self-attention (SA) and SA with KV cache to generate 256, 512, and 1024 tokens.

Table 6 reports the memory usage of each model. LNSSM maintains a constant memory footprint, independent of sequence length, thanks to its state design. In contrast, vanilla transformer consumes increasing memory as the sequence grows, due to the recomputation over the full context. KV-cached transformer, while reducing computational cost, requires additional memory to store all past key-value pairs, leading to higher peak memory usage than both LNSSM and standard transformers. These results highlight the efficiency advantage of LNSSM in both FLOPs and memory, making it particularly suitable for long-sequence generation under constrained hardware environments.

