# OpenReview forum: "Log-Normal State-Space Model"
_ICLR.cc/2026/Conference — ICLR 2026 Conference Withdrawn Submission_

### Official Review · Reviewer_6thV · 2025-10-25

**Soundness:** 2
**Presentation:** 2
**Contribution:** 2
**Rating:** 2
**Confidence:** 4

**Summary:**

The paper extends Linear Log-Normal Attention to SSM and validates its superiority compared to RWKV and Mamba.

**Strengths:**

The paper clearly explain the design of log-normal in SSM.

**Weaknesses:**

Experiments: No well-established Language Benchmark is tested. (in [lm eval harness](https://github.com/EleutherAI/lm-evaluation-harness)). Baselines are very limited: GLA, Gated Delta Networks, LongHorn are not included.

Writing: the motivation of this work is based on the "effectiveness" of log-normal distribution in self-attention. However, this is not a well-know conclusion and the paper does no provide any evidence for that. and conclusion is claimed inappropriately: "LNSSM matches the expressiveness of self-attention in transformers" -> what is expressiveness? and it is expected that your performance outperform Transformer if you claim this but no experiments compared to Transformer is done.

**Questions:**

refer to above weakness
1. Can you validate the LNSSM performance in lm eval harness and add at least one more baseline?
2. Can you show the "effectiveness" of log-normal distribution in attention like a Transformer with normal distributional attention is worse than the one with log-normal?

---

### Official Review · Reviewer_knEo · 2025-10-28

**Soundness:** 3
**Presentation:** 3
**Contribution:** 2
**Rating:** 4
**Confidence:** 3

**Summary:**

This paper introduces the Log-Normal State-Space Model (LNSSM), a novel architecture designed to bridge the performance gap between efficient State-Space Models (SSMs) and expressive self-attention mechanisms. The authors observe that the strong performance of modern SSMs (like Mamba and RWKV) relies not only on their time-mixing (recurrent) layers but also on intricately designed channel-mixing layers, which adds complexity and reduces interpretability. To address this, LNSSM leverages the theoretical insight that softmax attention weights approximately follow a log-normal distribution. By reformulating causal Linear Log-Normal (LLN) attention into a recurrent state-update form with an exponential feature map and a state decay mechanism inspired by RWKV, LNSSM aims to achieve the expressiveness of self-attention while maintaining the linear-time complexity and constant memory footprint of SSMs. Experimental results on language modeling (perplexity), question answering (BABILong), and long-form generation (ASQA, ELI5, WikiLarge) demonstrate that LNSSM outperforms several SSM baselines and remains robust with increasing sequence lengths, where standard transformers degrade.

**Strengths:**

* The paper is strengthened by a clear, critical observation: modern SSMs rely heavily on complex, specialized channel-mixing layers. The authors use this to motivate a highly controlled experimental setup, comparing all time-mixing methods using an identical, simple GPT-2 MLP. This elegantly isolates the variable of interest and proves that LNSSM's performance gains stem directly from its novel time-mixing core, not auxiliary architectural components. I quite like the methodology.

* By creating a time-mixing layer powerful enough to work with a standard MLP, LNSSM offers a path toward more efficient, interpretable, and expressive sequence models, addressing a central challenge in the field often overlooked by others.

**Weaknesses:**

* The comparison, while including strong baselines, could be more definitive. A comparison against the standard Mamba (or Mamba-2) block *with its original channel-mixing layer* is crucial to fully substantiate the claim that LNSSM achieves comparable performance with a simpler (GPT-2) MLP. The paper should state this crucial methodological choice explicitly and unambiguously in the main text of Section 5, not just bury it in a figure caption. This is by far my biggest concern.

* All experiments are conducted at a relatively small scale (~72M parameters). The conclusions would be significantly stronger if supported by results at a larger scale (e.g., 100M+ parameters) to ensure the findings hold as model size increases.

* The contribution of individual components in the overall LNSSM design (QK normalization, KV-shift, output gate, and most importantly, the state decay matrix A) is not systematically ablated. It is unclear how much each component contributes to the final performance.

* The extensive presentation of hand-derived gradient formulas and custom CUDA kernel algorithms in the appendix, while impressive from an engineering standpoint, is atypical for a core conference paper and distracts from the primary scientific contribution. A high-level summary of the implementation challenge and its solution would suffice. Note that I have *not* lowered my score because of this.

**Questions:**

* To directly validate the claim of reduced reliance on specialized channel-mixing, could you provide a performance comparison where LNSSM (with GPT-2 MLP) is directly compared against Mamba-2 with its original channel-mixing layer under identical settings and parameter budgets?

* Table 4 shows a dramatic speedup with a custom CUDA kernel. Since custom kernels are standard for performance I wanted to ask if there's anything specific to LNSSM that allows for such speedups?

---

### Official Review · Reviewer_hsxx · 2025-10-31

**Soundness:** 2
**Presentation:** 2
**Contribution:** 2
**Rating:** 2
**Confidence:** 4

**Summary:**

This paper introduces Log-Normal State-Space Models (LNSSM), a novel architecture that adapts the principles of Linear Log-Normal (LLN) attention for use within the State-Space Model (SSM) framework. LNSSM achieve superior performance on various downstream tasks compared to existing SSMs.

**Strengths:**

- Clarity and Readability: The paper is generally well-written, with a clear and logical flow that makes the core concepts accessible and easy to follow.

- Novelty of Idea: The central concept of mapping the log-normal attention mechanism from LLN to the SSM framework is novel and theoretically interesting, potentially opening a new avenue for SSM design.

- Empirical Results: The preliminary empirical results suggest that LNSSM is a promising approach, showing improved metrics over established baselines on the evaluated tasks.

**Weaknesses:**

While the core idea is interesting, the paper suffers from several major weaknesses in its current form, spanning methodological rigor, conceptual clarity, and overall structure.

- Structural Issues: The paper's organization hinders a clear understanding of its contributions.

  - It lacks a dedicated Related Works section, making it difficult to situate the LNSSM within the broader landscape of SSMs (e.g., S4, Mamba, RWKV) and related attention mechanisms.

  - The contributions are not explicitly enumerated.

  - A Limitations section is absent, which is crucial for a balanced scientific presentation.

  - Lack of line numbers

- Lack of Ablation Studies: In Section 4.5, the authors introduce several implementation choices (e.g., KV-shift) in the final architecture. It is impossible to tell if the reported performance gains come from the core LNSSM contribution or from these auxiliary design choices. The paper needs ablation studies to isolate the impact of LNSSM. For example:

  - What is the performance when replacing the LNSSM block with an S6 or Mamba block, while keeping all other architectural choices (like KV-shift) constant?

  - What is the performance impact of systematically removing these other design choices (KV-shift, etc.) from the final LNSSM model?

- Conceptual & Technical Clarity:

  - Parallelizability: Appendix A incorrectly implies that SSMs cannot operate in parallel. Models like S4 are well-known to be parallelizable as a long convolution (computed efficiently with FFTs) for training/prefill.  Can LNSSM be formulated in a parallel, non-recurrent mode for efficient training, similar to S4 and Mamba?

  - Comparison to LLN: The paper frames its contribution as applying a "mathematical trick" to LLN to achieve SSM properties (linear compute, constant state). However, it is unclear if the original LLN already possessed these properties. The paper must clarify the concrete differences in computational and memory complexity between LNSSM and the original LLN.

  - Figure 4 is disorganized and hard to follow. The symbol 'g' is undefined.

- Experiments: The experiments appear to be conducted at a relatively small scale. The ROUGE metrics in Table 3, for instance, are very low across all models, making it difficult to draw conclusions about practical effectiveness. The paper's claims would be substantially stronger if validated at a larger scale.

**Questions:**

- Parallelism: Does LNSSM support a parallel (convolutional) training mode for efficient prefill, similar to S4? If not, how does its training efficiency compare?

- Ablations: Can you provide ablation studies that isolate the performance gains of the core LNSSM block from other architectural choices like the KV-shift?

- LLN Comparison: What are the precise differences between LNSSM and original LLN? Furthermore, what are the computational and memory complexity differences between LNSSM and the original LLN? If they are similar, an experimental comparison on downstream performance of those two methods is needed.

- Figure 1: In Figure 1, were the causal convolution of Mamba and token shift of RWKV replaced for the comparison?

---

### Note · Authors · 2025-12-08

**Comment:**

We believe the current version of the paper requires further improvement and validation. Thus, we would like to withdraw it and resubmit in the future.

**Withdrawal Confirmation:**

I have read and agree with the venue's withdrawal policy on behalf of myself and my co-authors.